# Performance Comparison and Optimization of 16V265H Diesel Engine Fueled with Biodiesel Based on Miller Cycle

Feng Jiang [1,2], Junming Zhou [1], Jie Hu [1,2,*], Xueyou Tan [1], Qinglie Mo [1] and Wentong Cao [1]

[1] School of Mechanical and Automotive, Guangxi University of Science and Technology, Liuzhou 545006, China; 100001086@gxust.edu.cn (F.J.); a13078027926@163.com (J.Z.); tanxueyou123@163.com (X.T.); 13647728575@163.com (Q.M.); cao12580159@163.com (W.C.)

[2] Guangxi Key Laboratory of Automobile Components and Vehicle Technology, Guangxi University of Science and Technology, Liuzhou 545006, China

\* Correspondence: 100001078@gxust.edu.cn

**Abstract:** This paper introduces the theoretical basis and optimization method of diesel engine working process theory. By comparing two Miller cycle schemes of B20 biodiesel under different load conditions of 1000 rpm (100%, 75%, and 50%), the best Miller cycle scheme and the best Miller degree were found. Then, based on the Miller cycle scheme, its performance was optimized and analyzed, and the best intake timing scheme of the B20 biodiesel engine under different working conditions was obtained. The results show that the performance of B20 biodiesel in variable valve overlap angle Miller cycle is better than that in variable cam profile Miller cycle, and the effect is the best when the Miller degree is 30 °CA. When B20 biodiesel is used under 100% and 50% load conditions, the maximum power under the two loads is in the area near intake valve timing 179 °CAA and exhaust valve timing 174 °CAA, and intake valve timing 224.5 °CAA and exhaust valve timing 119 °CAA, respectively. Fuel consumption, soot emissions, and NOx emissions also have the corresponding best performance intake valve and exhaust valve positions.

**Keywords:** biodiesel; miller cycle; optimizer model



## 1. Introduction

At present, the transportation industry in various countries, especially the large-scale railway locomotive industry, is developing rapidly, and the demand for large-scale railway transportation is increasing in countries all over the world. That is why it is better to face the current market changes and demands and respond to the "carbon neutral" policy [1]. Innovatively burning low proportion biodiesel on a 16V265H diesel engine is the key technology to deal with the increasingly challenging energy problem and improve the comprehensive performance of this diesel engine [2], which has significantly improved the emission, economy, and power performance of locomotive diesel engines [3–5]. In this study, the working process of a 16V265H diesel engine fueled with biodiesel was simulated [6,7] and its performance was analyzed by two Miller cycle technologies [8]; through the optimization results of the optimization model formed by secondary development, the optimization of valve timing for the 16V265H diesel engine was studied under 100% and 50% operating conditions [9].

The power source (diesel engine) of a diesel locomotive in China adopts electronically controlled unit pump mode fuel injection [10,11], and this injection technology can achieve more accurate injection performance [12]. Based on the unchanged structure of the original complete machine and injection system, biofuel can effectively reduce the carbon monoxide, soot, and conventional and unconventional hydrocarbon emissions of diesel engines [13–17]. However, when diesel engines without any technical adjustment burn biodiesel, there is a certain negative impact on NOX emissions [18]. This is unmatched by traditional petrochemical diesels [19,20] and it is therefore necessary to conduct basic

research on the low ratio biodiesel of locomotive diesel engines. A low ratio means that the volume content of biodiesel in the mixed fuel is low. For example, B20 is defined as 20% of the volume content of biodiesel in the mixed fuel.

The Miller cycle is a process of reducing the actual compression ratio of the engine by closing the intake valve in advance so that the actual mixture entering the cylinder is less than the theoretical value. In this way, the expansion ratio will be greater than the compression ratio, thus improving the combustion efficiency [21–23]. At the same time, the Miller cycle is also closely related to its duration. Closing the inlet valve too early will reduce the working medium and the pressure in the cylinder, but it is not conducive to the improvement of thermal efficiency, which will worsen the internal aerodynamic loss and increase the fuel consumption. There are two ways to realize Miller's cycle. One is to change the intake cam profile, keep the intake valve opening time unchanged, and advance the intake valve closing time; that is, change the intake valve lift curve, also known as variable cam profile [24]. The other is to change the installation angle of the intake cam to achieve the purpose of the overlap angle of the variable valve. The opening and closing time of the intake valve will change at the same time, which is called the overlap angle of the variable valve [25–27]. Figure 1 below shows the comparison between the Miller cycle and the conventional diesel engine cycle indicator diagram, and further explains the reason why the Miller cycle reduces $NO_X$ emission. The Miller cycle has closed the intake valve at an angle of 9′ before the bottom dead center. As the piston moves downwards, the gas entering the cylinder will get an additional expansion process, which will produce the effect of cooling the gas in the cylinder. Therefore, Miller cycle technology is more advanced in controlling combustion temperature [28,29]. The effect of this temperature reduction will continue throughout the combustion process. Therefore, $NO_X$ emission will also decrease with the decrease in combustion temperature [30–34]. As shown in the figure, curve 8-1-2-3-4-5-6-7-8 is the cycle process of a traditional diesel engine. The intake stroke is curve 8-1, the compression stroke is curve 1-2, the combustion expansion stroke is curved 2-3-4-5, and the exhaust stroke is curved 5-6-7. The Miller cycle process is curved 8′-9′-1′-2′-3′-4′-5′-6′-7′-8′. Among them, the intake stroke is curve 8′-9′, the compression stroke is 1′-2′, the combustion expansion stroke is 2′-3′-4′-5′, and the exhaust stroke is curve 5′-6′-7′. It can be seen from the cycle indicator diagram that the intake pressure of the Miller cycle is greater than that of the ordinary diesel engine cycle; however, the peak pressure in the cylinder is equal to the circulating pressure of an ordinary diesel engine.

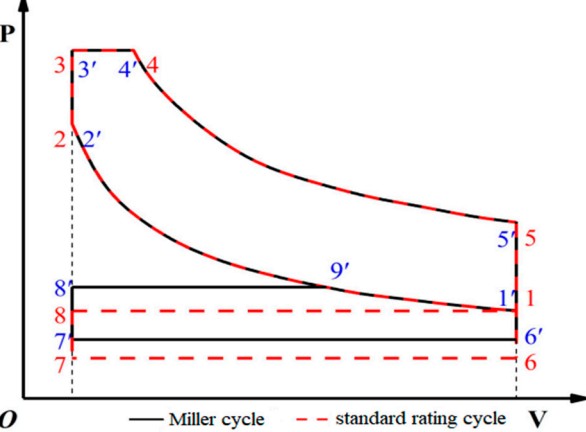

**Figure 1.** Comparison between Miller cycle and traditional diesel engine cycle indicator diagram.

At present, scholars at home and abroad have begun to study the use of biodiesel in large locomotive diesels and the application of Miller cycle technology. Roper Edward et al. [35] conducted confirmatory simulation using Ricardo Wave and studied the power and emission performance of diesel engines based on the Miller cycle and low ratio carbon fuel. The results show that the perfect combination of the two is an excellent scheme to reduce

carbon, nitrogen emissions, and particulate matter. Wei et al. [36] studied the combination of exhaust gas recirculation (EGR) and the Miller cycle to reduce the loss of engine fuel economy. The results show that the high fuel consumption can be compensated by reducing the pumping loss under the corresponding load conditions. Allami et al. [37] compared the changes in engine combustion (engine performance) and exhaust emission by using the injection system of a 6-cylinder diesel engine running at a constant speed of 1500 rpm with biodiesel of B10, B25, and B40 in different concentrations ratios. Hasan et al. [38] studied in detail the various aspects of compression ignition engines fueled with methanol/diesel fuel. The results show that the mixing ratio of diesel and methanol will change under different load conditions. The mixing ratio directly affects the $NO_X$ emission the higher the proportion of mixed methanol and the higher the content of $NO_X$. Tong et al. [39] studied the effects of Cyclopentanone and diesel blending on the combustion and emission of common rail diesel engines. The experimental results show that biodiesel with a 20% concentration ratio has the best effect on reducing particulate and $NO_X$ emissions at an 8% EGR rate and has great application potential in diesel vehicles.

　　In this paper, the simulation calculation method is used to study the combustion and emission characteristics, power, and economy of biodiesel. Biodiesel is a kind of high-quality clean diesel [40–43], which can be extracted from various biomass, and so it can be said to be inexhaustible energy. Against the background of resource depletion, researchers in various countries have taken it as a substitute for ordinary diesel [44–47]. This paper analyzes B20 concentration biodiesel and focuses on the degree of influence on engine power, economy, and emission under this concentration ratio. Through calculation and test, the performance index of a locomotive diesel engine without changing the structure when burning biodiesel was compared with the Miller cycle characteristics of the 16V265H diesel engine under three different working conditions when using B20 concentration ratio biodiesel. Based on the above results, the optimizer model was used to optimize the intake and exhaust timing of the 16V265H diesel engine with the maximum power, the best fuel consumption, and the lowest emission as the optimization objectives, The optimal valve timing scheme of a B20 biodiesel-fueled diesel engine under 100% and 50% operating conditions was obtained.

## 2. Numerical Approaches

### 2.1. Theory of Diesel Engine Working Process

　　This paper studies the working process of a 16V265H diesel engine and simulates it through the modules in GT-Power simulation software. The basic differential equation of the working process in the cylinder is the core equation in the whole simulation calculation, and using this equation it can be applied to the calculation of other models such as the combustion model and emission model. Based on this workload, the convergence of the flow and pressure coefficients of the diesel engine can be obtained through the results of the above differential equations. It is necessary to simplify the working process (as a thermodynamic system), describe it with a mathematical model, and establish a conservation equation. The following assumptions shall be made before model calculation:

(1) The gas in the cylinder is ideal, and there is no leakage during the sealing process. The characteristic values such as specific enthalpy $h$, specific internal energy $u$, and specific heat capacity $C$ are only related to the gas temperature $T$ and gas composition;

(2) The state of the working medium in the cylinder of the diesel engine is the same everywhere (i.e., the pressure, temperature, and concentration of the working medium), and the gas left in the previous cycle should be fully mixed with the fresh air charge in the cylinder at the boundary of the intake stroke of this cycle;

(3) The kinetic energy of the fresh working medium during the cycle stroke, the temperature and pressure changes in the fresh working medium during the intake process are ignored, and the flow process of the fresh working medium is quasi-stable.

　　Based on the above assumptions, three basic parameters (cylinder pressure: $P$; temperature: $T$; and mass: $m$) are used to describe the working medium state in the diesel engine

cylinder during the working process. The whole cylinder working process is related to the energy conservation equation, mass conservation equation, and ideal gas state equation.

(1)　Energy conservation equation [48]:

$$\frac{dU}{d\varphi} = h_s \frac{dm_s}{d\varphi} + \frac{dW_Q}{d\varphi} - \frac{dm_e}{d\varphi} h_e - \frac{dW_R}{d\varphi} - p\frac{dV}{d\varphi} \tag{1}$$

where:

　　$U$—System internal energy, kJ;
　　$W_Q$—Heat energy generated by fuel combustion inside the cylinder, kJ;
　　$W_R$—Heat energy exchanged by modules of cylinder system, kJ;
　　$h_s$—Common mass-specific enthalpy of the inlet valve;
　　$h_e$—Common mass-specific enthalpy of the exhaust valve.

(2)　Mass conservation equation:

Obtained from the principle of conservation of mass [49], the total mass of exchange at the model boundary in the thermodynamic system is equivalent to the mass change of the working medium; that is, the following equation:

$$\sum_j dm_j = dm \tag{2}$$

The total exchange mass at the boundary of the thermal system model without mass leakage is:

$$\frac{dm}{d\varphi} = \frac{dm_s}{d\varphi} + \frac{dm_e}{d\varphi} + \frac{dm_B}{d\varphi} \tag{3}$$

where:

　　$m$—Internal working medium quality of cylinder;
　　$m_e$—Cylinder outflow waste mass;
　　$m_B$—Mass of fuel burned in the cylinder.

(3)　Equation of state of an ideal gas:

$$pV = mRT \tag{4}$$

where:

　　$m$—Mass of working medium in the cylinder, kg;
　　$V$—The working volume of a cylinder, m$^3$;
　　$p$—Working medium pressure in the cylinder, Pa;
　　$T$—Temperature of working medium in the cylinder, K; $R$—Gas constant.

### 2.2. Theoretical Basis of Diesel Engine Application Optimization Method

Whether in the field of engineering technology or the field of economy, the optimization of key objectives is very important in the decision-making process where the so-called optimization is to make the best choice among different alternatives. If the alternative is recorded as a design variable, and the measurement index to measure its quality is defined by the objective function, a typical optimization problem can be described by the following mathematical problems:

$$\min f(x) \tag{5}$$

$$s.t.\ h_i(x) = 0, i = 1, 2\,, \mathrm{m} \tag{6}$$

$$g_i(x) \geq 0, j = 1, \ldots, p \tag{7}$$

where *f(x)* is the objective function; *X* is the n-dimensional vector (design variable); Equation (6) is the constraint condition of equality; and Equation (7) is the constraint condition of inequality. *X* satisfying Equations (6) and (7) are called the feasible solution of the optimization problem, and the set of all feasible solutions constitutes the feasible

region. As the solution to the optimization problem, its essence is to find the extreme value of the objective function *f(x)* under the two constraints of Formulas (6) and (7). It is to find the overall optimal solution in the feasible region. The subsequent optimization problems involved in this paper can also be expressed in the above form:

$$\min f(\theta_1, \theta_2) \tag{8}$$

$$s.t.\ \theta_{10}\ \leq\ \theta_1\ \leq\ \theta_{11} \tag{9}$$

$$\theta_{20}\ \leq\ \theta_2\ \leq\ \theta_{21} \tag{10}$$

Among them, the optimization variables $\theta_1$, and $\theta_2$ are the intake and exhaust timing angles, and the two variables are the design variables. The optimization objective function *f(x)* can be set as the physical quantities that vary linearly with the random variables, such as maximum power or minimum fuel consumption, and the constraint condition is inequality constraint; $\theta_{10}$, $\theta_{11}$ are the lower and upper limits of the value range of intake valve timing variables, respectively, while $\theta_{20}$, $\theta_{21}$ are the lower and upper limits of the exhaust valve variable value range.

At present, optimization design is still a very active research field. According to the types of the objective function and constraint function, it can be divided into a linear optimization problem and a nonlinear optimization problem [50].

Most of the linear optimization problems are solved by designing the simplex table with the simplex method [51]. When studying the nonlinear multi-objective function problems, if the function is constrained, it can be solved by approximating the quadratic polynomial near the extreme point of the objective function through the Taylor expansion. If the function is unary, it can be solved by the one-bit search golden section method [52]. If the function is a multivariate one, the problem can be transformed into a set of nonlinear equations according to the value conditions of the extreme points of the multivariate function. The available methods are the steepest descent method, Newton method, conjugate gradient method, etc.

For a wider range of nonlinearly constrained optimization problems in the engineering field, the usual optimization methods can be divided into two categories: analytical method (indirect method) and direct method. The core of the direct method is to limit the original objective function to the feasible region for search, and approach the target value step by step in the search process until the search approaches an optimal solution in the feasible region.

Further, if the optimization constraints condition is a linear function of the design variable and the objective function is a convex function of the design variable, this kind of optimization problem can be expressed as the following convex programming problem:

$$\min f(x), x \in R^n \tag{11}$$

$$s.t.\ c_i\left(x\right) = a_i^T x + b_i = 0, i \in E = \{1, \ldots, l\} \tag{12}$$

$$c_i(x)\ \geq\ 0, i\ \in\ I = \{1+l, \ldots, m\} \tag{13}$$

(1)   $x^*$ is the local optimal solution of the above problem, and the effective set is $I^* = \{i\,|\,c_i(x^*) = 0, i = 1, 2, \ldots, m\}$;
(2)   *f(x)*, $c_i(x)$ ($1 \leq I \leq M$ is differentiable at point $x^*$);
(3)   For $i \in I^*$, $\nabla c_i(x^*)$ is linearly independent. Then there is vector $\lambda^* = \{\lambda_1^*, \lambda_2^*, \ldots, \lambda_n^*\}$.

$$\nabla f(x*) - \sum_{i=1}^{n} \lambda_i^* \nabla ci(x*) = 0, \lambda_i^* ci(x*) = 0, i = 1, \lambda, m\ \lambda_i^* \geq 0, i = 1, \lambda\ m \tag{14}$$

Such a point is the KT point, and in this kind of problem, if the design vector $x^*$ is the KT point of the problem, then $x^*$ is the optimal solution to the problem.

As shown in Equations (11)–(13), and according to the following requirements for the objective function in the feasible region, the optimization problem involved in this paper can be regarded as a convex programming problem.

In this model, the real-time variable (RLT) along with the design variable is the objective function of the optimization model. An appropriate iterative method is selected to obtain the optimal design vector sequence through its iteration and finally, a series of objective function sequences are obtained. When we want to get the optimal solution, then the objective function sequence is convergent.

Generally speaking, in the optimization model, it is required that the variation curve of the optimization objective function within the feasible region of the design variable is shown in Figure 2a, to ensure that there is a unique optimal solution for the optimization objective within the feasible region. If the variation law curve of the optimization daily standard function with the design variable is shown in Figure 2b, the value of the optimization objective function is nonunique, and the iteration step of the design variable needs to be further refined so that the optimization objective function still presents the characteristics of a single extreme point convex function within the iteration step. As shown in Figure 2c, reduce the value range of design variable parameters and divide the value range of design variables into four interval ranges A1-A2-A3-A4-A5. Carry out local optimization design in each interval, compare and confirm the relevant objective function values, and finally determine that the optimal solution is in the range of intervals A3-A4. If the range of the A3-A4 design interval is further subdivided into B1-B2-B3-B4-B5, it can be determined that the optimization solution is within the range of b4-b5. This idea can make the objective function approach the optimal solution step by step.

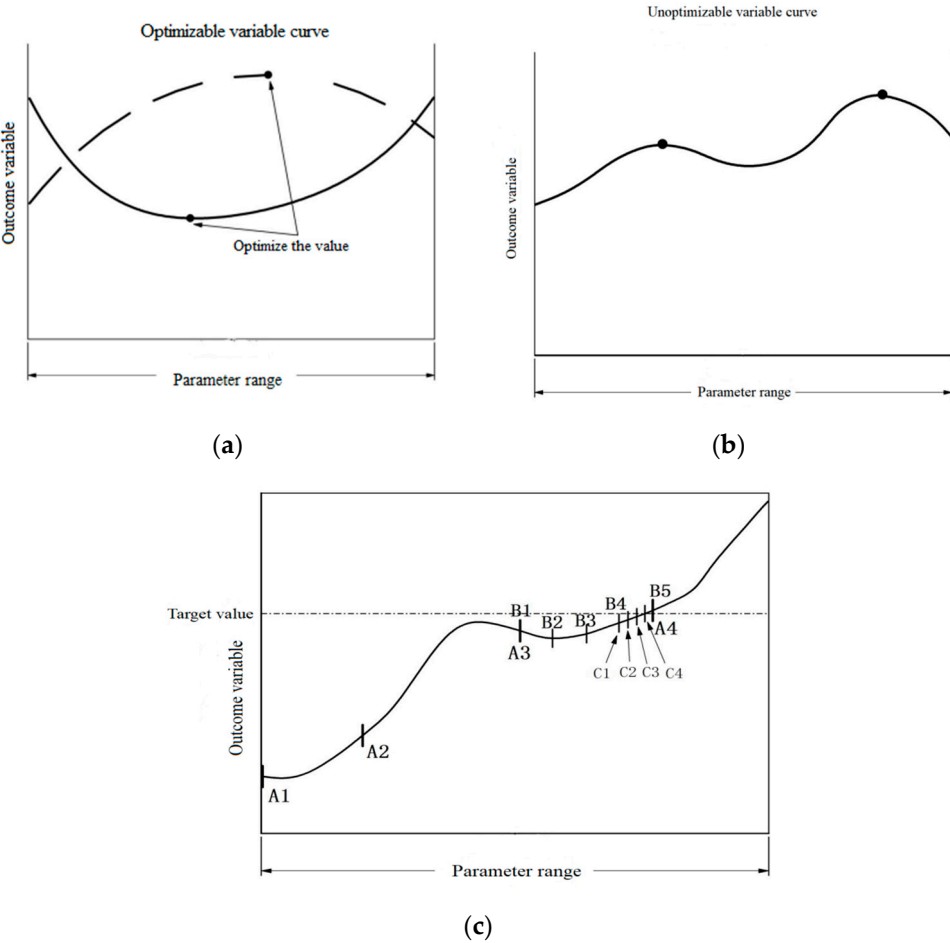

**Figure 2.** Function mode between RLT variable and design variable of optimization objective function. (**a**) Optimizable variable curve. (**b**) Nonoptimizable variable curve. (**c**) Full range variable curve.

After determining that the change trend curve of the optimization design variables and the objective function are as shown in Figure 2a, the optimization objective function is regarded as a univariate function in each iteration step using coordinate rotation, and the Brent method is selected for search and solution.

With the diesel engine power $P_t$ as the reference value and the external factors defined, $P_t$ can be expressed as a function of the intake and exhaust valve timing angles $\varphi_Q$ and $\varphi_W$. The results show that the power also varies with the timing angle of the intake and exhaust valves. To facilitate the subsequent calculation and reduce the amount of calculation, the mathematical model can be expressed as:

$$\min P_t = f_1(x), x = \left[\varphi_Q, \varphi_W\right]^T \tag{15}$$

$$st.x(1) \in [149.5, 269.5] \tag{16}$$

$$x(2) \in [76, 196] \tag{17}$$

In the same way, the mathematical model related to fuel consumption and emissions can be derived.

Derived from optimization principles, after the initial search point $x$ is determined, the direction in which the function value drops fastest is the negative gradient direction, so the new search point is determined as:

$$\nabla f_1(x_0)\beta_0 + x_1 = x_0 \tag{18}$$

Among them, when $\beta_0$ takes different values, a new search point will be generated, resulting in corresponding changes in the function value of $f_1(x_1)$.

$f_1(x_1)$ changes with the change in positive real number $\beta_0$, that is, $f_1(x_1) = \beta_0$. To get the minimum value of $f_1(x)$ we need to find out the known values of all functions and take their extreme values to get:

$$h(\beta_0) = \frac{\partial \psi_1(\beta_0)}{\partial \beta_0} = 0 \tag{19}$$

To calculate the value of equation 19, the Brent method is often used by GT-Power software to calculate the value of $\beta_0$.

In the optimization calculation of this study, there are two variables: intake timing angle and exhaust timing angle. The problem can be classified into constrained optimization problems and solved by the Brent method.

### 2.3. Overall Simulation Model

Figure 3 shows the simulation model of the 16V265H diesel engine. The 16V265H diesel engine power assembly includes cylinder, piston, connecting rod, valve assembly, valve, intake and exhaust cam, etc. The cylinder model setting mainly includes the cylinder's basic dimension and cylinder boundary conditions (including the cylinder head, cylinder wall, piston temperature distribution, etc.), in-cylinder combustion model, and heat transfer model. Input the known basic dimension parameters of the cylinder in the cylinder geometry module, and the initial conditions of the cylinder are set according to the original parameters. GT-Power has various combustion models for the engine. In this study, the diesel engine adopts the combustion model "EngcCylCombDIJet" with predictable functions for simulation calculation, and the Woschni classic heat exchange model is selected in the heat transfer module [53–55].

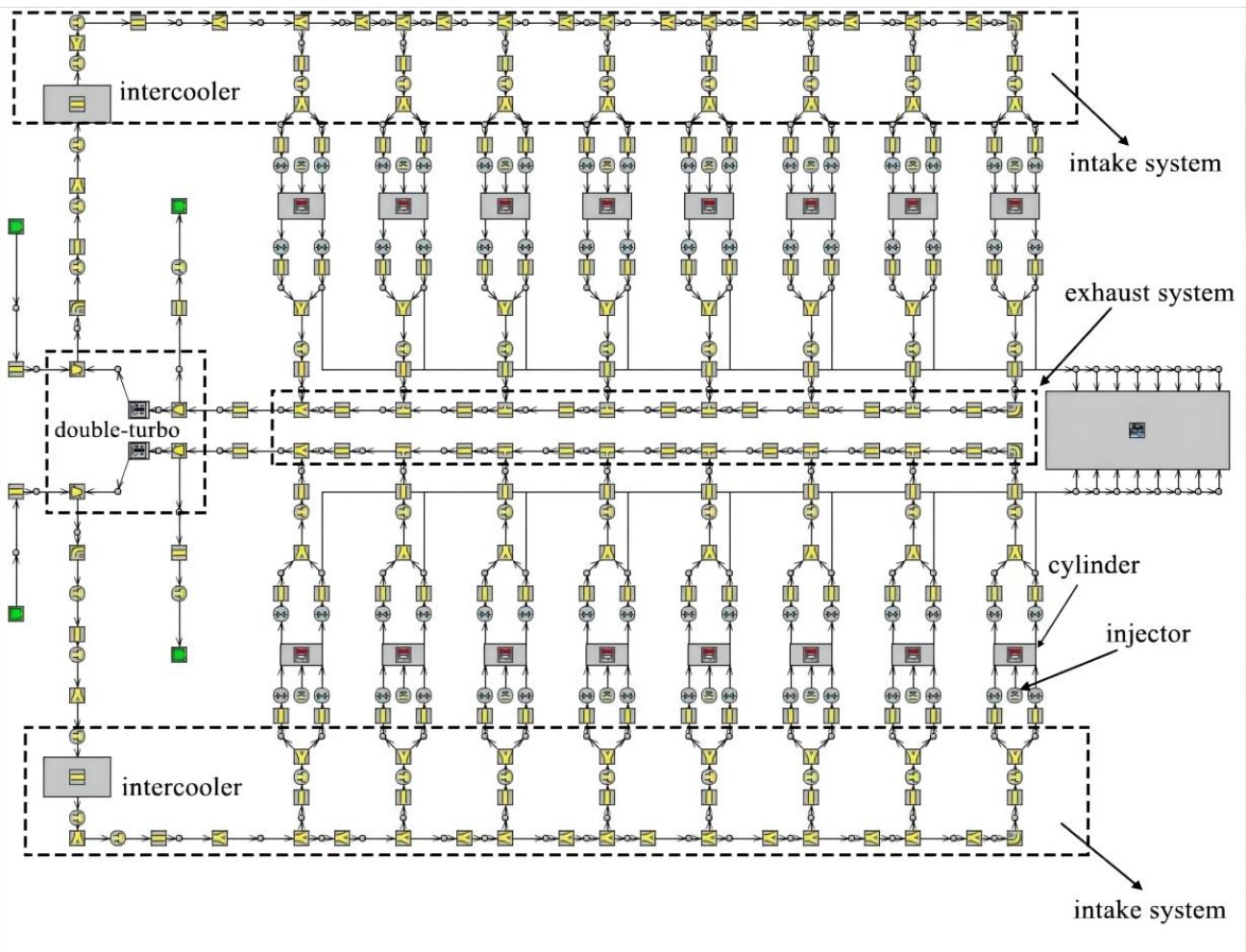

**Figure 3.** Simulation model of the 16V265H diesel engine.

## 3. Results and Discussion

### 3.1. Two Miller Cycle Principles

In this research and analysis, two different methods are used to realize the Miller cycle. The first method is to change the intake convex profile, keep the intake valve opening time unchanged, and advance the intake valve closing time; that is, change the intake valve lift curve, also known as variable cam profile, which will be replaced by VCP later. The second method is to change the installation angle of the intake cam to achieve the purpose of valve overlap angle. The opening and closing time of the intake valve will change at the same time, which is called the valve overlap angle, and will be replaced by VVA later.

In this section, two Miller cycles are applied to the 16V265H diesel engine to study its performance characteristics under the conditions of B20 biodiesel at 50%, 75%, and 100% load. The load condition refers to the corresponding change in throttle opening with the change in output power (load). For example, in 50% and 75% load conditions; that is, the throttle opening is within the range of 50~75% and the diesel engine works under this load condition most of the time. At this time, the mixture with economic concentration is supplied to ensure the engine has better fuel economy. The 100% load condition is also called the full load condition; that is, the throttle valve approaches or reaches the fully open position. At this time, the diesel engine is required to produce maximum power to overcome large external resistance or accelerate driving. Based on the valve lift curve of the original engine the VCA and the VVA are adopted and seven schemes for early closing of inlet valves are designed, as shown in Table 1.

**Table 1.** Early closing of inlet valve in two schemes.

| Miller Degree (°CA) | 0 | 10 | 20 | 30 | 40 | 50 | 60 | 70 |
|---|---|---|---|---|---|---|---|---|
| Translation angle of intake valve lift curve (°CA) | 0 | 10 | 20 | 30 | 40 | 50 | 60 | 70 |
| Intake valve closing time (°CA) | 540 | 530 | 520 | 510 | 500 | 490 | 480 | 470 |

After determining two Miller cycle schemes, GT-Power is used to establish a calculation model to study the performance characteristics of a 16V265H diesel engine under two Miller cycles. The calculation process should ensure that the fuel injection amount of the Miller cycle is the same as that of the original engine cycle. In addition, the diesel engine should increase the boost pressure during the Miller cycle to ensure sufficient air intake. Otherwise, the diesel engine combustion will deteriorate due to insufficient air intake and the fuel combustion will be insufficient resulting in increased $NO_X$ and soot emissions. Figure 4 shows two Miller cycle (variable valve lift) schemes.

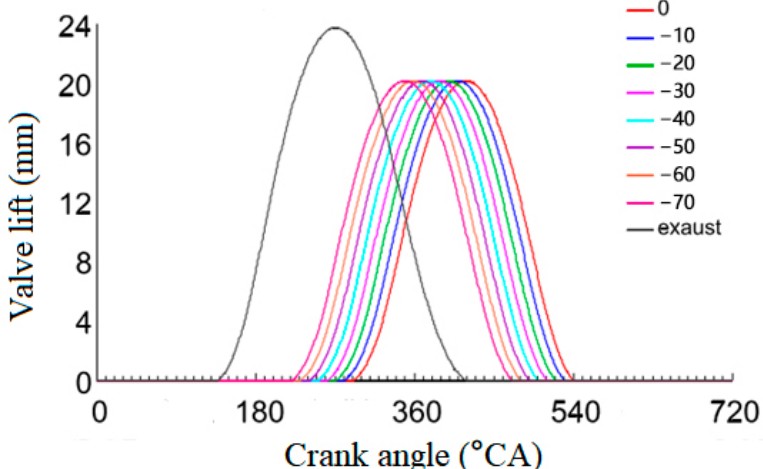

**Figure 4.** The curve of variable valve lift.

*3.2. Performance Comparison of B20 Biodiesel in Two Miller Cycle Modes*

3.2.1. Comparative Analysis of Two Miller Cycle Power of B20 Biodiesel under Different Loads

Figure 5 shows the power diagram of VCP and VVA using B20 biodiesel under 100%, 75%, and 50% load conditions.

According to the data in the figure, when the diesel engine is under 100% load conditions the power of VCP and VVA shows an upward trend within the range of 0–30 °CA, and within this range the power of VCP is slightly less than that of VVA. The reason is that the intake volume of the VCA cylinder will be lower than VVA when Miller degree is in the range of 0–30 °CA, and the VCP will fuse the fuel and gas more fully in the cylinder so that the combustion will be more thorough and the power will be greater. Between 0–30 °CA, the maximum lift of the inlet valve becomes shorter and the intake duration decreases compared with the VVA, while the lift of VVA remains unchanged. At this time, the valve overlap angle is very small, which hinders the flow of fresh air and thins the air volume in the exhaust pipe. When the Miller degree is in the 40–70 °CA range, the power values of the two show the opposite. The power value of the VCP is greater than that of the VVA, and there is an obvious downward trend in this range. The reason is that the air intake of this section of VCP gradually increases, while the intake air volume of the VVA in the 40~70 °CA section is significantly smaller than that in the 0~30 °CA section. The decrease in intake air volume leads to insufficient combustion and power decline of the diesel engine and its decreasing range is much larger than the power of VCP. Therefore, the inflection point of the two Miller cycles concerning power at 100% load is 30 °CA.

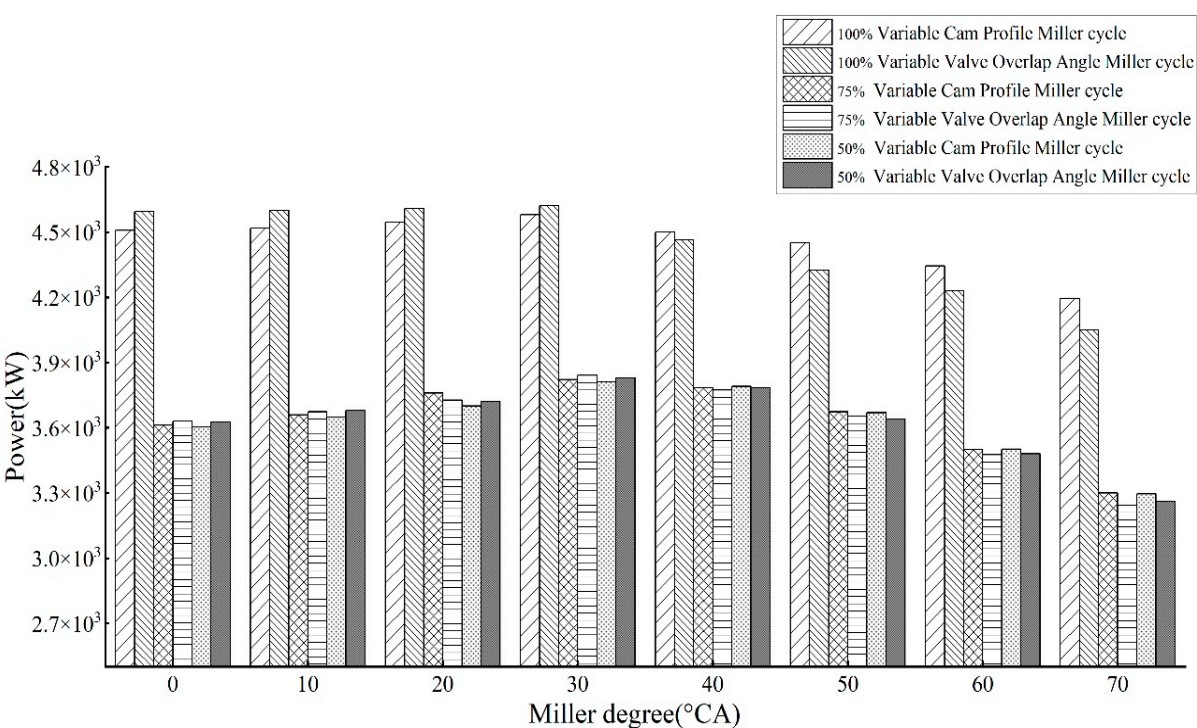

**Figure 5.** Power analysis under three working conditions.

Similarly, under 75% and 50% load conditions, the two Miller cycle powers also show an upward trend in the range of 0~30 °CA, and within this range, the power of VCP is slightly less than that of VVA. This is consistent with the 100% load condition, but the power difference between the two Miller cycles at 75% and 50% load is slight compared with the 100% load condition. Under 100% load, the maximum difference between the two Miller cycle powers is about 145 kw (Miller degree 70 °CA), and the minimum difference is about 35 kW (Miller degree 40 °CA). At 75% load, the maximum difference between the two Miller cycle powers is about 79 kW (Miller degree 70 °CA), and the minimum difference is about 6 kW (Miller degree 40 °CA). At 50% load, the maximum difference between the two Miller cycle powers is about 36 kw (Miller degree 70 °CA), and the minimum difference is about 12 kW (Miller degree 40 °CA). When the Miller degree is within the range of 40~70 °CA, the power of VCP under 75% load and 50% load is slightly higher than that of VVA, which is similar to that under 100% load. Within this range, both showed a significant downward trend and the reasons for both power reductions are the same as at 100% load. Therefore, at 75% and 50% load the inflection point of two Miller cycles relative to power is also 30 °CA.

3.2.2. Comparative Analysis of Fuel Consumption of Two Miller Cycles of B20 Biodiesel under Different Loads

Figure 6 shows the fuel consumption diagram of VCP and VVA using B20 biodiesel under 100%, 75%, and 50% load conditions.

According to the figure, when the diesel engine is at 100% load, the fuel consumption of VCP and VVA shows a downward trend within the range of 0–20 °CA; the reasons for this phenomenon are that the air intake of the two Miller cycles is increased within this range, the air and fuel are fully integrated, and the combustion is more thorough. It can also be seen from the figure that the fuel consumption of the VCP is slightly greater than that of the VVA within this range. When the Miller degree is greater than 20 °CA the fuel consumption of the two cycles begins to increase, and when the Miller degree is gradually greater than 30 °CA the fuel consumption of VCP is lower than VVA. This is because the air intake of the VCP increases continuously and a large amount of fresh air is

obtained in the cylinder, which has an endothermic cooling effect on the cylinder; for the VVA, after the Miller degree is 30 °CA the intake air volume decreases continuously and the oil–gas mixture is not sufficient. To ensure that the power is not affected the circulating fuel injection volume can only be increased, so the fuel consumption increases. Therefore, at 100% load, the inflection point of the two Miller cycles relative to fuel consumption is 20 °CA.

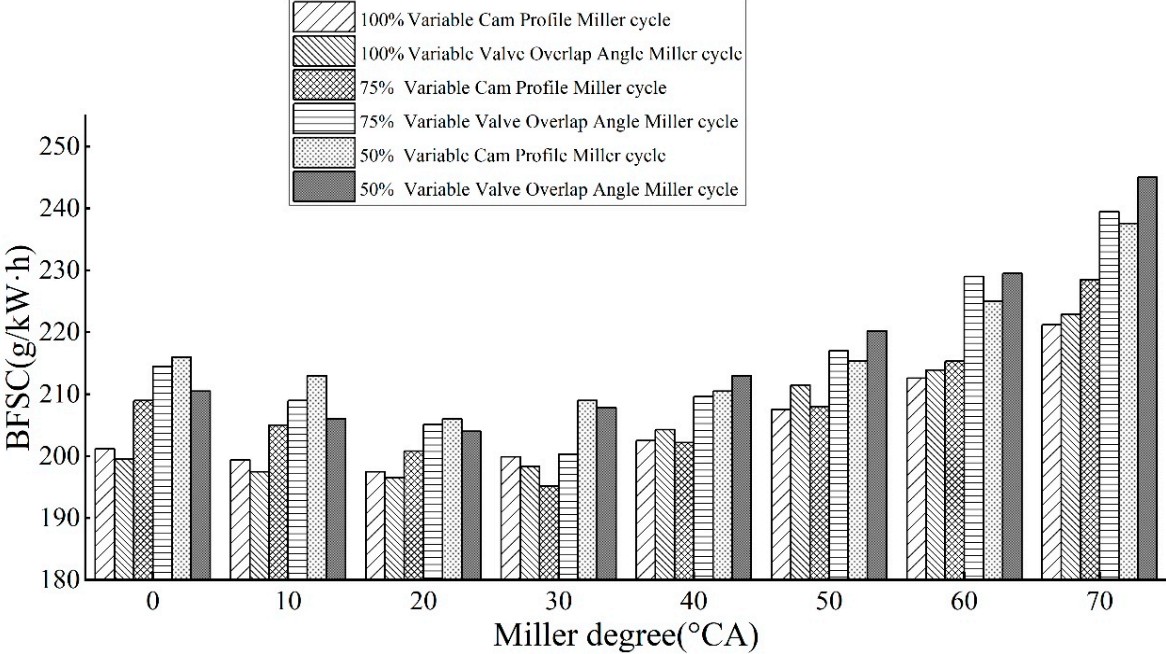

**Figure 6.** Fuel consumption analysis under three working conditions.

Figure 6 also shows that the fuel consumption of the VCP and the VVA shows a downward trend in the range of 0–30 ° CA under 75% load, and the fuel consumption of the VCP is significantly lower than that of the VVA. When the Miller degree is within the range of 40–70 °CA the fuel consumption of the two Miller cycles shows a sharp rise, but the rising rate of the VCP is lower than the VVA and the fuel consumption is still lower than the VVA. The reason for this is that when the Miller degree is greater than 30 °CA the intake air is reduced and the oil–gas mixture of B20 biodiesel is insufficient. To obtain sufficient power, the diesel engine needs to increase the amount of circulating fuel injection; that is, the fuel consumption increases. Therefore, at 75% load the inflection point of the two Miller cycles relative to fuel consumption is 30 °CA.

Under 50% load conditions, the fuel consumption of VCP and VVA shows a downward trend within the range of 0–20 °CA and the fuel consumption of the VCP is higher than that of the VVA. When the Miller degree is within the range of 20–30 °CA the data begin to rise. When the Miller degree is within the range of 30–40 °CA the fuel consumption of VCP is gradually lower than VVA, and after 40 °CA the fuel consumption of VCP is completely lower than VVA. Therefore, at 50% load the inflection point of fuel consumption of two Miller cycles is 20 °CA. This is roughly similar to the experimental results of Guan et al. [56].

3.2.3. Comparative Analysis of Soot Emission from Two Miller Cycles of B20 Biodiesel under Different Loads

Figure 7 shows the soot emission diagram of VCP and VVA using B20 biodiesel under 100%, 75%, and 50% load conditions.

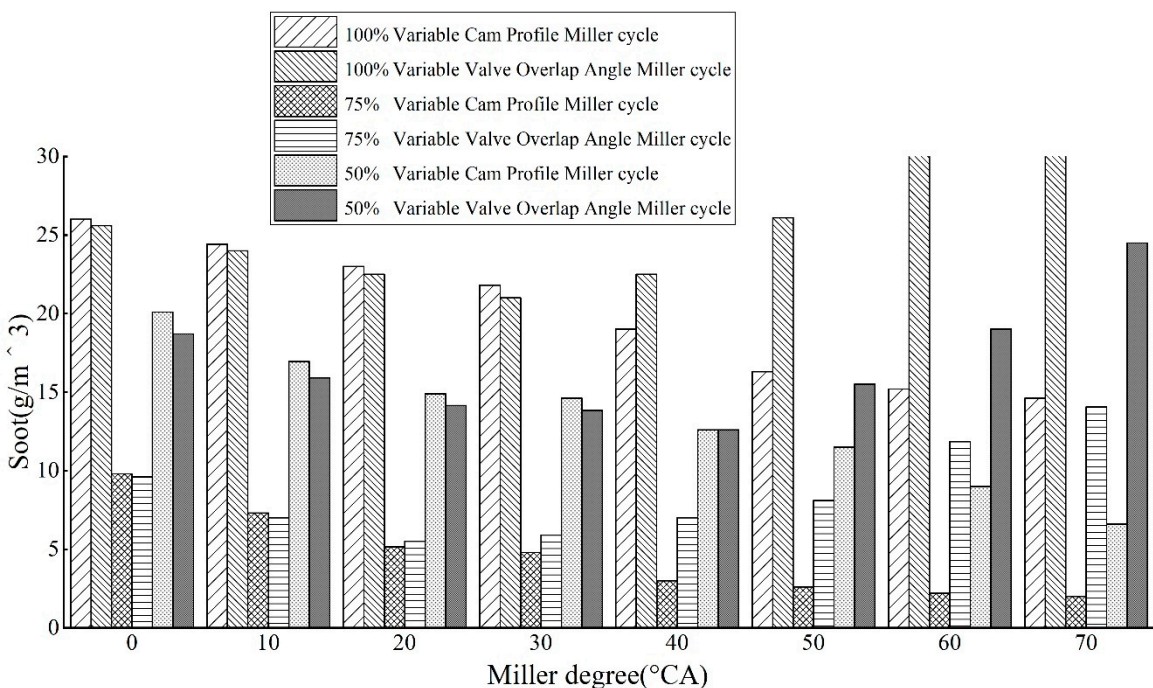

**Figure 7.** Soot emission analysis under three working conditions.

According to the figure, when the diesel engine is under 100% load conditions the fuel consumption of VCP and VVA shows a downward trend within the range of 0–30 °CA, and the VCP is slightly higher than the VVA but the difference is almost the same. The reason is that the inlet pressure of two Miller cycles increases, the fuel is completely burned, and the soot emission is reduced. When the Miller degree is in the range of 30–70 °CA the soot emission of the VVA has a sharp upward trend, while the soot emission of the VCP continues to maintain a downward trend increasing the difference between the two. The reason is that the overlap angle of the VVA increases, and part of the pressurized incremental air flows out of the exhaust channel reducing the air intake; however, the air intake of the VCP continues to increase, and the oil and air are fully mixed, the combustion in the cylinder is more thorough, and the soot emission is gradually reduced. Therefore, at 100% load, the inflection point of the two Miller cycles relative to soot emission is 30 °CA.

Figure 7 also shows that the soot emission of the VCP and the VVA shows a downward trend in the range of 0–20 °CA under 75% load, and the soot emission values are almost the same for both cycles; the reasons for this are the same as those under the above 100% load conditions. When the Miller degree is within the range of 20–70 °CA the smoke and dust emissions of the VVA begin to show an upward trend, especially when the Miller degree is 50 °CA and the rising rate increases significantly. The reason is that when the Miller degree increases, part of the pressurized incremental air flows out of the exhaust channel resulting in the reduced intake of air, insufficient combustion, and increased smoke and dust emission. In sharp contrast, when the Miller degree increases the intake of VCP increases and this further reduces soot emissions. Therefore, at 75% load the inflection point of the two Miller cycles burning B20 biodiesel relative to soot emission is 20 °CA.

Under 50% load condition the soot emission of VCP and VVA shows a downward trend in the range of 0–40 °CA and the VCP is higher than the VVA. When the Miller degree is 40 °CA the two are almost the same. The increase in Miller degree causes the increase in intake air volume, which makes B20 biodiesel fully burned in the cylinder and reduces soot emission. Within this range the VVA has a better soot emission effect. When the Miller degree is within the range of 40–70 °CA, the soot emission of the VVA begins to rise sharply. However, the soot emission of the VCP continues to decrease due to the

continuous increase in air intake within this range. Therefore at 50% load, the inflection point of two Miller cycles burning B20 biodiesel relative to soot emission is 40 °CA.

### 3.2.4. Comparative Analysis of NO$_X$ Emission of B20 Biodiesel in Two Miller Cycles under Different Loads

Figure 8 shows the NO$_X$ emission diagram of VCP and VVA using B20 biodiesel under 100%, 75%, and 50% load conditions.

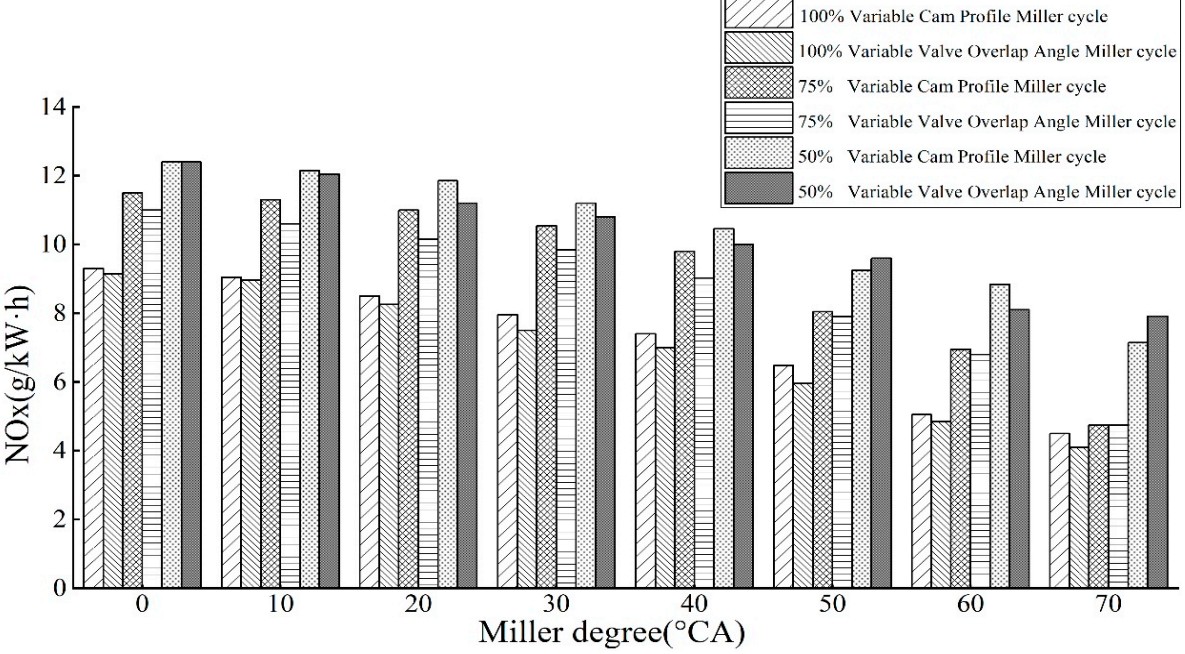

**Figure 8.** NO$_X$ emission analysis under three working conditions.

According to the figure, when the diesel engine is under 100% load conditions the NO$_X$ emissions of VCP and VVA show a downward trend within the range of 0–70 °CA. At 50 °CA the maximum difference between the two is 0.53 g/kW·h. It can be concluded that the VVA is significantly better than the VCP in terms of NO$_X$ emission, and the reasons for the reduction of NO$_X$ emission are not the same. In the VCP, when the Miller degree increases fresh air will go through an additional expansion process when entering the cylinder and will accompany the whole combustion process. The existence of fresh air will reduce the temperature in the cylinder due to cooling and thus reduce NO$_X$ emissions. In the VVA with the increase in Miller degree the intake pressure will increase, and the low-pressure gas in the cylinder will not be easily discharged during the exhaust stroke; this is equivalent to that in the high-pressure intake that will compress part of the exhaust in the cylinder, resulting in the continuous increase in exhaust gas content and the reduction of air content (especially oxygen content) thereby reducing NOx emissions.

Figure 8 also shows that NO$_X$ emission of VCP and VVA shows a downward trend in the range of 0–70 °CA under 75% load, which is the same as that under 100% load conditions. Under this working condition, when the Miller degree is 70 °CA the NO$_X$ emissions produced by the two Miller cycles tend to be equal. Due to the formation mechanism of NO$_X$ the combustion state is "high-temperature oxygen enrichment", the VCP intake continues to increase, the combustion is more complete, and the temperature in the cylinder is higher. The high-pressure inlet of VVA compresses part of the exhaust gas into the cylinder to reduce the oxygen content in the cylinder and keep the temperature in the cylinder at a low level; therefore, the NO$_X$ emission from the VCP is higher than that from the VVA. Because the reasons for the reduction of NO$_X$ emissions are different, when the Miller degree increases the factors affecting the air intake of the two Miller cycles become increasingly similar. Therefore, the NO$_X$ emissions from the combustion of the

two Miller cycles above 50 °CA are closer. From the analysis it can be seen that the $NO_X$ emissions from the VVA combustion are lower than those from the VCP combustion, and their performance is better.

Under 50% load condition the $NO_X$ emission of the VCP and the VVA shows a downward trend within the range of 0–70 °CA, the value of the VCP is significantly higher than that of the VVA, and the reasons for the decrease in $NO_X$ are the same as that under 100% and 75% load conditions. Under this working condition, when the Miller degree is 50 °CA, the $NO_X$ emissions produced by the two are similar. When the diesel engine is fueled with B20 biodiesel and the Miller degree is 70 °CA, the $NO_X$ emission produced by the VCP is lower than that produced by the VVA; this is because the generation of $NO_X$ emissions is related to the temperature and oxygen content in the cylinder. It can be concluded that the combustion effect of VVA is better than that of VCP when B20 biodiesel is burned under this working condition. This is roughly similar to the experimental results of Wang et al. [57].

### 3.3. Optimization Analysis of B20 Biodiesel under Different Loads

Based on the two Miller cycle optimization schemes, the intake and exhaust timing of the 16V265H diesel engine can be used as the independent variable of the optimization model. Based on the optimizer model and taking the maximum or minimum value of each result variable as the optimization objective, calculate the intake and exhaust timing (do not change the valve lift, only optimize the timing angle) and compare the performance indicators before and after optimization. At this speed, the optimal valve timing under 100% and 50% load conditions is determined, which improves the performance of the diesel engine.

#### 3.3.1. Power Optimization Analysis of B20 Biodiesel under Different Loads

Figures 9 and 10, respectively, show the maximum power optimization calculation results of the 16V265H diesel engine fueled with B20 biodiesel under different load conditions (100%, 50%) at 1000 rpm. It can be seen from the following effect cloud picture that when the red area tends to be concentrated, the calculation results will be closer to the optimization target value. The abscissa of the cloud diagram represents the independent variable intake valve timing angle, and the ordinate represents the exhaust valve timing angle of the independent variable. Each point distributed in the cloud diagram represents the valve timing angle corresponding to a certain power value. The more points distributed in the red area, the better the area is.

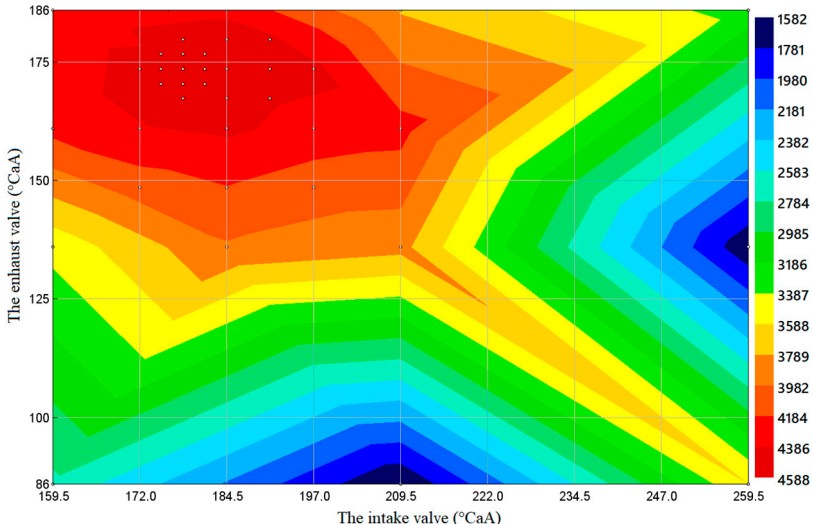

**Figure 9.** 100% load condition.

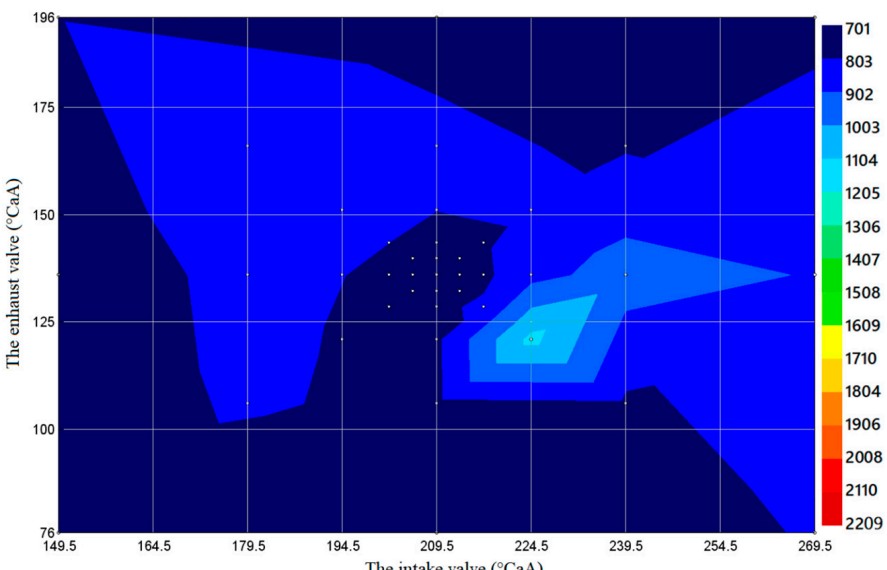

**Figure 10.** 50% load condition.

Figure 9 shows that the maximum power optimization dense distribution area appears near the inlet valve timing 172–190 °CAA and the exhaust valve timing 165–180 °CAA: The main reason is that within the valve timing range the intake volume of fresh air corresponding to the valve will increase, the oil and gas in the diesel engine cylinder will be fully integrated, the combustion will be more thorough, and the power will increase accordingly. With approaching intake valve timing 209.5 °CAA (initial intake valve timing angle), approaching exhaust valve timing 86–90 °CAA, approaching intake valve timing 259.5 °CAA, and exhaust valve timing 137 °CAA, the power is very small. The reason is that when the diesel engine is fueled with B20 biodiesel at full load the corresponding valve overlap angle in this area reduces the intake volume in the cylinder, the mixing effect of B20 biodiesel and the air is poor, and the power decreases. According to the optimization results, close to the intake valve timing 179 °CAA and the exhaust valve timing 174 °CAA, there is an optimal optimization region; that is, there is a maximum output power region.

Figure 10 shows that the maximum power area is very small, and the maximum value appears near 224.5 °CAA of intake valve timing and 119 °CAA of exhaust valve timing. It can be seen from the cloud diagram that the effect of the power optimization value generated by the diesel engine burning B20 biodiesel under this working condition fails to meet the expectations. This is because the intake airflow in the cylinder of the diesel engine is less under the timing valve angle, the B20 biodiesel fuel and air are not fully mixed, and the thermal efficiency and power of the diesel engine are reduced. According to the results, near 224.5 °CAA of intake valve timing and 119 °CAA of exhaust valve timing the output power of the diesel engine is the largest.

Figure 11 shows the power comparison before and after optimization of the 16V265H diesel engine burning B20 biodiesel at 100% and 50% load conditions at 1000 rpm with the maximum power as the optimization result. The figure shows that after optimization the power value increases slightly under both load conditions, but it is not significant with the increase in the value. This is because the diesel engine can burn more fully under two load conditions, so the change in output power difference is small. However, this scheme can achieve maximum power and improve the dynamic performance of the locomotive, which has certain practical significance for this diesel engine burning B20 biodiesel.

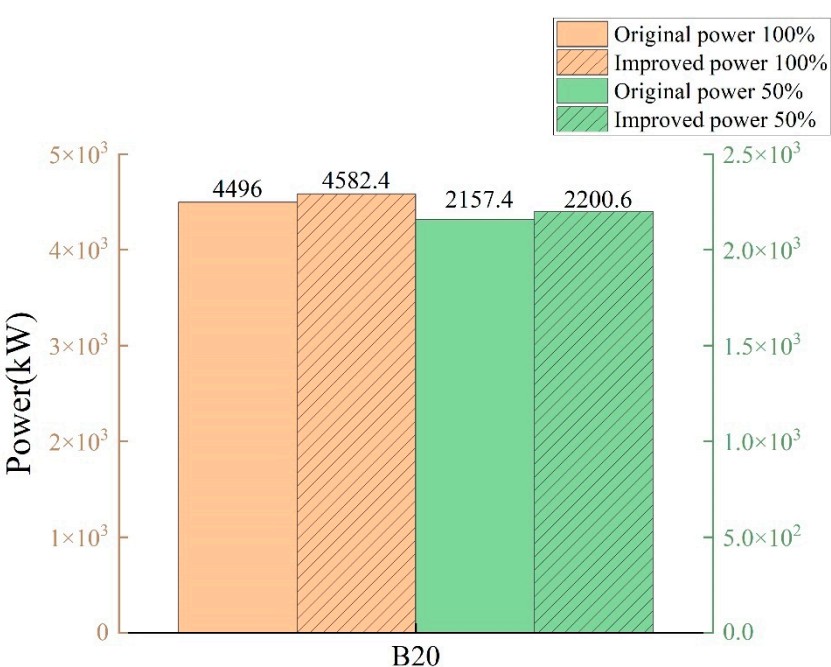

**Figure 11.** Comparison between the improved power and the original value under 100% and 50% load.

### 3.3.2. Fuel Consumption Optimization Analysis of B20 Biodiesel under Different Loads

Figures 12 and 13 show, respectively, the minimum fuel consumption optimization calculation results of the 16V265H diesel engine fueled with B20 biodiesel under different load conditions (100%, 50%) at 1000 rpm. It can be seen from the cloud diagram that the more concentrated the blue area is the closer the calculated value is to the optimization target value, and some optimized valve timing value distribution points are concentrated around the blue area; the abscissa of the cloud diagram represents the independent variable intake valve timing angle. The ordinate represents the independent variable exhaust valve timing angle.

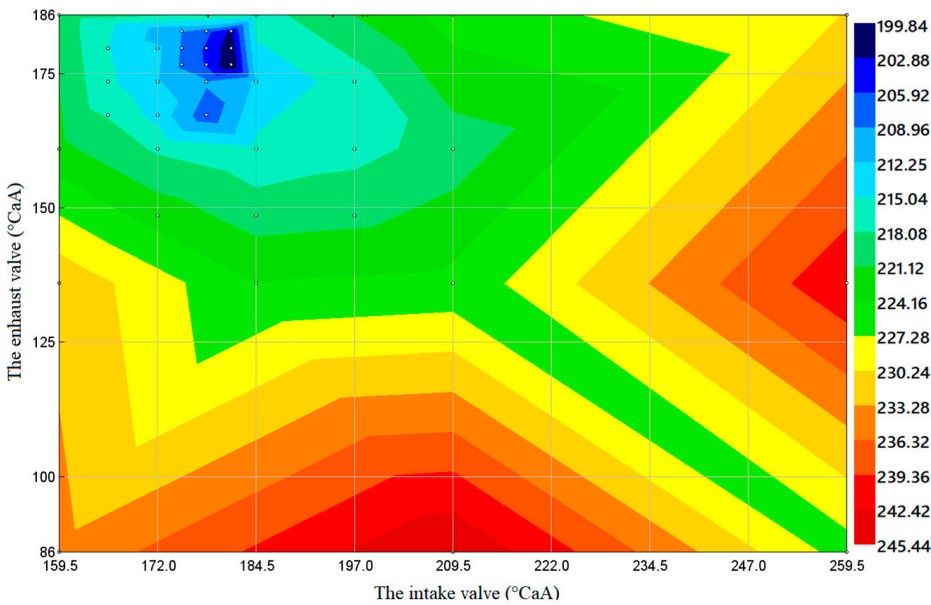

**Figure 12.** 100% load condition.

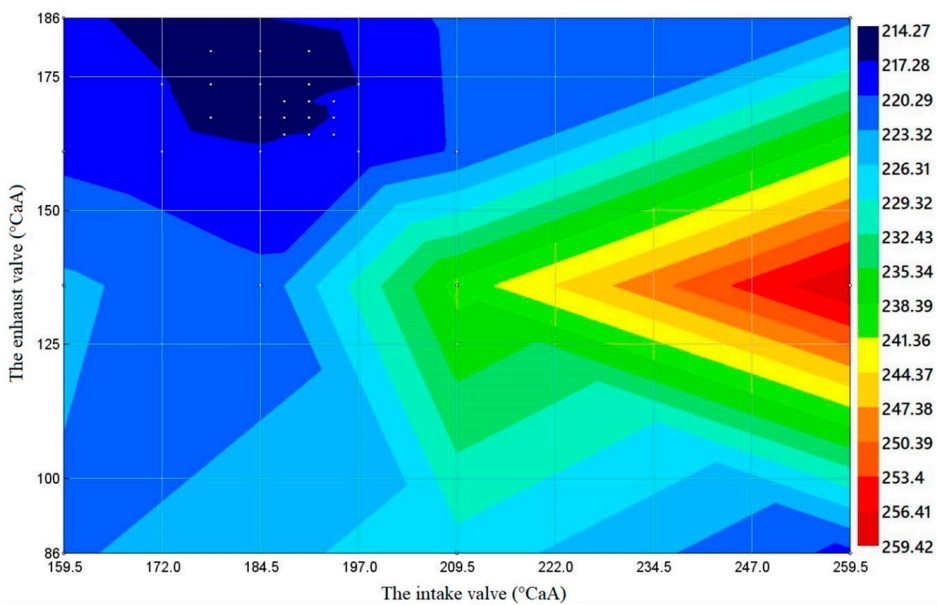

**Figure 13.** 50% load condition.

Figure 12 shows that there is a small area with dense distribution of minimum fuel consumption optimization near the intake valve timing 180 °CAA and exhaust valve timing 180 °CAA. It can be determined that under 100% load conditions the fuel consumption of the diesel engine is the lowest when the intake valve closing delay is about 29 °CAA and the exhaust valve opening is about 44 °CAA in advance (near the intake valve timing 180 °CAA and the exhaust valve timing 180 °CAA).

Figure 13 shows that when the diesel engine is fueled with B20 biodiesel, there are small areas with dense distribution of minimum fuel consumption optimization near 187.5 °CAA of intake valve timing and 170 °CAA of exhaust valve timing (when the intake valve closes needs to be delayed by about 22 °CAA and the exhaust valve opening needs to be advanced by about 34 °CAA). The reason is that when the valve is timing the amount of air caused by its overlap angle increases, which makes the fuel mix fully, the combustion more complete, and the fuel consumption lower. In the area near 259.5 °CAA of intake valve timing and 136.5 °CAA of exhaust valve timing is the area with maximum fuel consumption. The reason is that when the diesel engine uses B20 biodiesel at 50% load, the valve overlap angle corresponding to the region makes the intake airflow in the cylinder less, and the mixing effect of B20 biodiesel fuel and the air is poor so the fuel consumption increases.

Figure 14 shows the fuel consumption comparison before and after optimization of the 16V265H diesel engine burning B20 biodiesel at 100% and 50% load at 1000 rpm and taking the optimal fuel consumption as the optimization result. The figure shows that after the optimization of the optimizer model, the fuel consumption of the diesel engine under two load conditions is slightly lower than that before the optimization. This is because the diesel engine uses B20 biodiesel and air to fully mix and burn after optimization, which reduces the fuel consumption. Under 50% load conditions the decrease range reaches 4.54%, which plays a guiding role in the economic performance of the diesel engine.

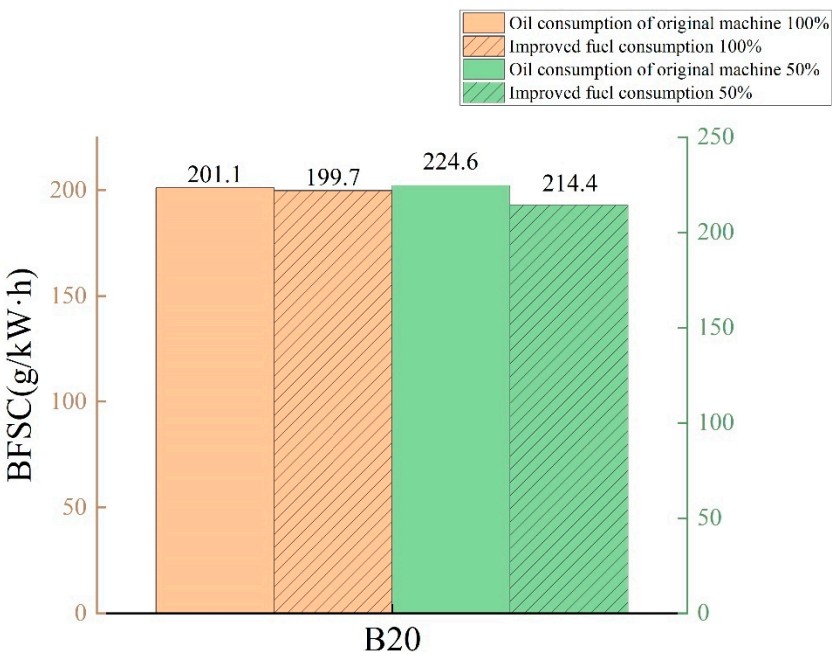

**Figure 14.** Comparison of improved fuel consumption with original value at 100% and 50% load.

### 3.3.3. Soot Optimization Analysis of B20 Biodiesel under Different Loads

Figures 15 and 16 show the optimization results of soot emission under three different load conditions (100%, 50%) at 1000 rpm when the diesel engine is fueled with B20 biodiesel. In the cloud image rendering, the more concentrated the red area the greater the smoke emission in the area. The more concentrated the blue area and quantity distribution, the area is closer to the optimal target value.

Figure 15 shows the optimization results of soot emission under 1000 rpm speed and 100% load when the diesel engine was fueled with B20 biodiesel. According to the cloud image, the maximum and minimum areas of soot emission appear in the areas near the inlet valve timing 239.5 °CAA and the exhaust valve timing 136.5 °CAA, and the areas related to the inlet valve timing 264.5 °CAA and the exhaust valve timing 191 °CAA, respectively.

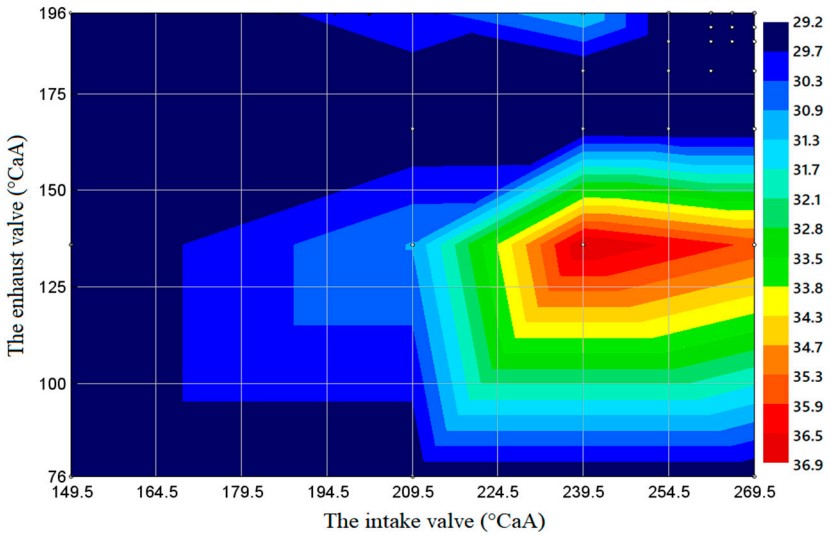

**Figure 15.** 100% load condition.

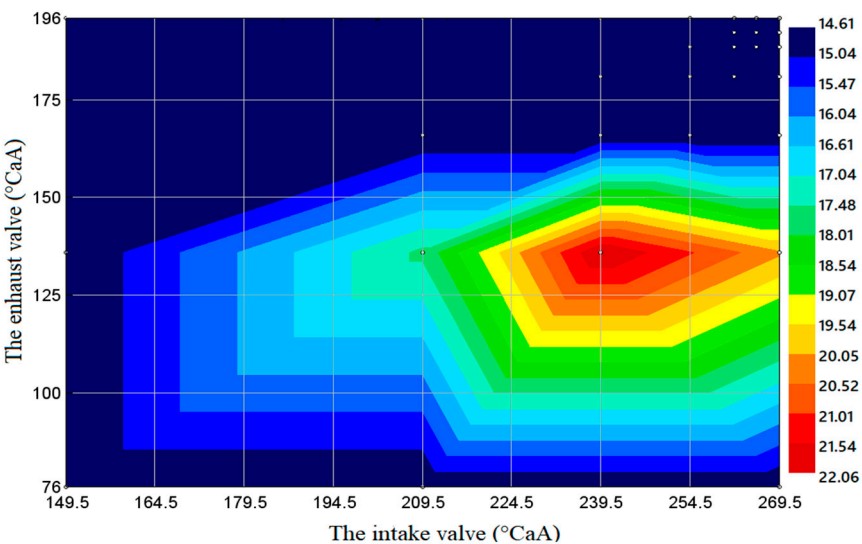

**Figure 16.** 50% load condition.

Figure 16 shows the optimization results of soot emission under 1000 rpm speed and 50% load when the diesel engine was fueled with B20 biodiesel. In the area around the inlet valve timing 239.5 °CAA and the exhaust valve timing 136 °CAA, the area with the maximum soot emission was mainly due to the minimum intake of valve overlap angle near the valve timing, incomplete fuel combustion, and the maximum soot emission.

Figure 17 shows the emission comparison before and after optimization of the 16V265H diesel engine burning B20 biodiesel at 100% and 50% load at 1000 rpm and taking the lowest soot emission as the optimization result. The figure shows that after the optimization of the model, the soot emission of the diesel engine under 100% and 50% working conditions was lower than that before the optimization; compared with 100% load condition, the soot emission after 50% load condition optimization decreases more significantly because the air intake condition is good, and the combustion is sufficient under this load condition. The optimization method has practical significance for diesel engine emission performance.

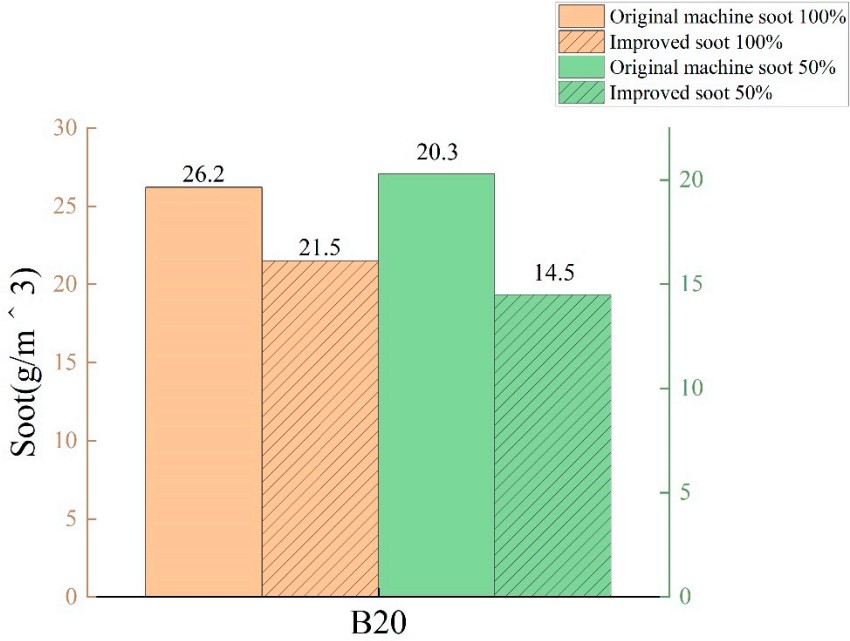

**Figure 17.** Comparison of improved soot emissions with original values at 100% and 50% load.

### 3.3.4. NO$_X$ Optimization Analysis of B20 Biodiesel under Different Loads

Figures 18 and 19 show the optimization results of NO$_X$ emission at 1000 rpm under two different load conditions (100%, 50%) when the diesel engine was fueled with B20 biodiesel. In cloud image rendering, the more concentrated the red area the greater the NO$_X$ emission in the area. The more concentrated the blue area and quantity distribution, it means that this area is the best performing area for NOx emission.

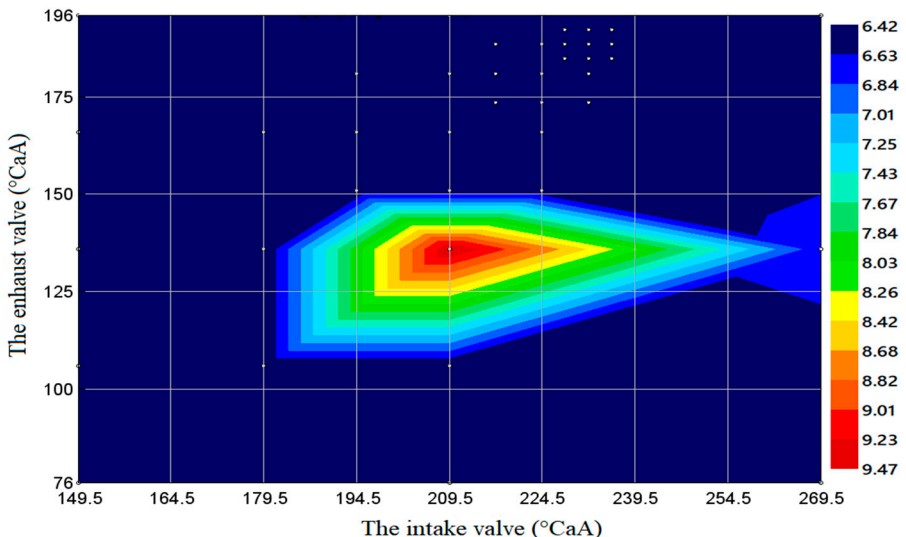

**Figure 18.** 100% load condition.

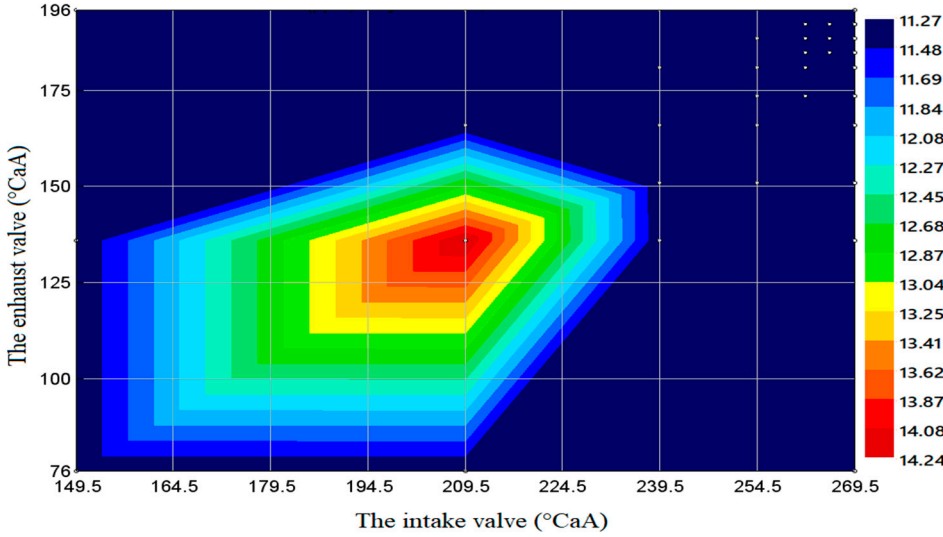

**Figure 19.** 50% load condition.

Figure 18 shows the optimization results of NO$_X$ emission at 1000 rpm speed and 100% load when the diesel engine was fueled with B20 biodiesel; the area with the maximum NO$_X$ emission occurs in the area around the intake valve timing 209.5 °CAA and the exhaust valve timing 136 °CAA. The lowest NO$_X$ emission area appears in the area centered around 232 °CAA of intake valve timing and 189 °CAA of exhaust valve timing.

Figure 19 shows the optimization results of NO$_X$ emission at 1000 rpm speed and 50% load when the diesel engine was fueled with B20 biodiesel; the area with the maximum NO$_X$ emission occurs in the area around the intake valve timing at 209.5 °CAA and the exhaust valve timing at 136 °CAA. The area around 264.5 °CAA of intake valve timing and 188 °CAA of exhaust valve timing is the area with the minimum NO$_X$ emission.

Figure 20 shows the emission comparison before and after optimization of the 16V265H burning B20 biodiesel at 100% and 50% load and 1000 rpm with the minimum NOx emission as the optimization result. According to the cloud diagram, the NOx emission under the two load conditions has been improved after optimization and the effect is good. This optimization method has practical significance for the emission performance of diesel engines.

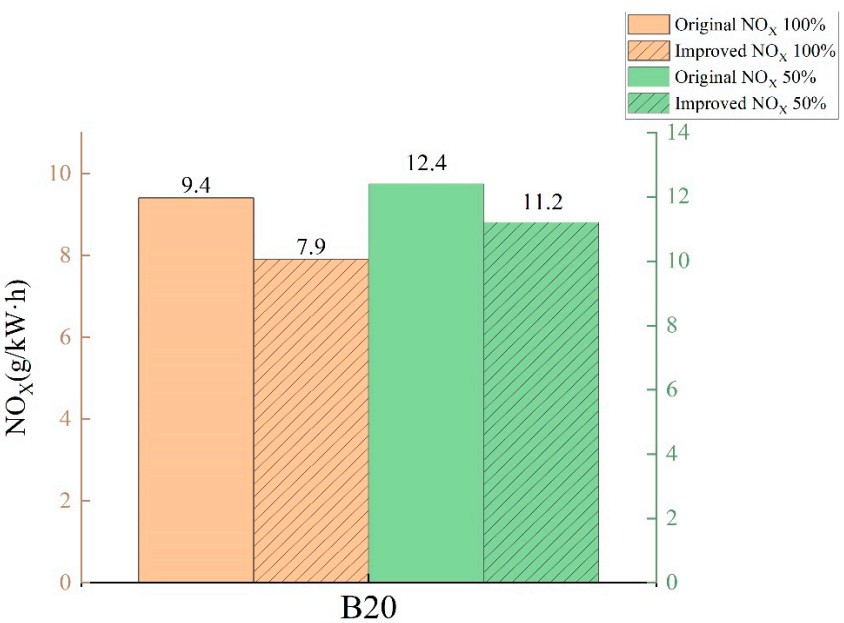

**Figure 20.** Comparison of improved $NO_X$ emissions with original values at 100% and 50% load.

## 4. Conclusions

The biodiesel has been shown to improve the combustion and emission characteristics of marine dual fuel diesel engine [58–60], which is of great significance to reducing carbon emission [61–64]. Firstly, this paper studied the influence of two Miller cycle technologies on the performance of power, fuel consumption, $NO_X$, and soot emission of the diesel engine when burning B20 biodiesel at 1000 rpm and under different load conditions and selects the appropriate Miller cycle mode. Subsequently and based on the optimizer optimization model, the diesel engine fueled with B20 biodiesel was studied under different load conditions taking the parameters such as diesel power, fuel consumption, and emission as the optimization objectives. The main conclusions are as follows:

(1) When the Miller degree increases to 0–30 °CA, the power of the VCA is lower than that of the VVA and the fuel consumption and emission are slightly higher than that of the VVA. When the Miller degree continues to increase to the range of 40–70 °CA, the power generated by both Miller cycles decreases at the same time and the former decreases less than the latter; the former has lower fuel consumption and soot than the latter and has better performance in fuel consumption and soot emission, but the $NO_X$ emission is slightly higher than the latter;

(2) It can be seen from the analysis that the effect of the VVA on improving the comprehensive performance of the locomotive diesel engines is significantly better than the VCA. From the optimization results of various performance indexes under 100%, 75%, and 50% load conditions, the best optimization scheme combination of a diesel engine can be determined as follows: the effect of improving the performance of diesel engines by using the VVA and Miller degree 30 °CA is the best;

(3) When using B20 biodiesel under 100% load conditions, when the intake valve closing needs to be delayed by about 30 °CAA and the exhaust valve opening needs to be advanced by about 28 °CAA (near the intake valve timing 179 °CAA and the exhaust valve timing 174 °CAA), the output power of the diesel engine has a maximum area.

When the intake valve closing is delayed by about 29 °CAA and the exhaust valve opening is advanced by about 44 °CAA (near the intake valve timing 180 °CAA and the exhaust valve timing 180 °CAA), the fuel consumption is the lowest. The soot emission is the lowest in the area around 264.5 °CAA of intake valve timing and 191 °CAA of exhaust valve timing. Near 232 °CAA of intake valve timing and 189 °CAA of exhaust valve timing is the area where $NO_X$ emission has a minimum value.

(4) When using B20 biodiesel under 50% load condition the diesel engine has the maximum output power when the intake valve closing needs to be delayed by about 15 °CAA and the exhaust valve opening needs to be advanced by about 17 °CAA (near 224.5 °CAA of intake valve timing and 119 °CAA of exhaust valve timing). Near 187.5 °CAA of intake valve timing and 170 °CAA of exhaust valve timing (when intake valve closing needs to be delayed by about 22 °CAA and exhaust valve opening needs to be advanced by about 34 °CAA), the lowest fuel consumption area appears. The area around 239.5 °CAA of intake valve timing and 136 °CAA of exhaust valve timing is the area with maximum soot emission. The lowest $NO_X$ emission area appears in the area centered around 202 °CAA of intake valve timing and 98 °CAA of exhaust valve timing.

**Author Contributions:** Conceptualization, F.J.; formal analysis, F.J., J.H. and J.Z.; software, F.J. and J.Z.; investigation, F.J., W.C., J.H., X.T., Q.M. and J.Z.; resources, F.J.; writing—original draft preparation, F.J. and J.Z.; writing—review and editing, F.J., W.C., J.Z., X.T., Q.M. and J.H.; supervision, F.J.; funding acquisition, F.J. and J.H. All authors have read and agreed to the published version of the manuscript.

**Funding:** This research was funded by the Doctoral Fund Project of Guangxi University of Science and Technology, grant number 21Z34 and 21Z46. The research also was funded by Independent research project of Guangxi Key Laboratory of Automobile Components and Vehicle Technology, grant number 2022GKLACVTZZ02 and 2022GKLACVTZZ03.

**Institutional Review Board Statement:** Not Applicable.

**Informed Consent Statement:** Not Applicable.

**Data Availability Statement:** All data used to support the findings of this study are included within the article.

**Acknowledgments:** The authors acknowledge the Doctoral Fund Project of Guangxi University of Science and Technology and Independent research project of Guangxi Key Laboratory of Automobile Components and Vehicle Technology.

**Conflicts of Interest:** The authors declare no conflict of interest.

## Nomenclature

| | |
|---|---|
| B20 | 80% diesel + 20% biodiesel |
| $NO_x$ | Nitrogen Oxides |
| BTE | Brake Thermal Efficiency |
| BSFC | Brake Specific Fuel Consumption |
| EGR | Exhaust Gas Re-circulation |
| DFP | Davidon–Fletcher–Powell Algorithm |
| RLT | Reformulation-Linearization-Technique |
| VCP | Variable Cam Profile Miller Cycle |
| VVA | Variable Valve Overlap Angle Miller Cycle |
| CO | Carbon Monoxide |
| BFSC | Brake Specific Fuel Consumption |

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
