# Peer review of "Performance Comparison and Optimization of 16V265H Diesel Engine Fueled with Biodiesel Based on Miller Cycle"

_processes, doi:10.3390/pr10071412_

Round 1

Reviewer 1 Report

The authors updated the paper very well. Explanations have been made in all key sections of the paper. I do not have some major issues with the paper. 

Reviewer 2 Report

The manuscript has improved tremendously from its previous submission. Based on its technical content and scientific soundness, the manuscript can now be accepted for publication, but I will leave the English and formatting to the editing team to work with the authors.

Reviewer 3 Report

·      The manuscript contains clear research background, sufficient literature review and acceptable simulation method.

·      However, the results and discussion section lacks conciseness and produces the same intonation/writing styles at different optimization conditions. It is suggested that the authors rewrite the discussion with more interesting perspectives, not just spelling out the trend and its corresponding reason, repetitively.

·      In abstract, the author should briefly mention the problem statement. Also, the writings from “When biodiesel is used at 100%, ... “ until the end of abstract is too long, too illustrative and very uncomfortable to read. It is suggested to revise it into a more concise statement(s) of research findings.

·      In 3rd row of Introduction, please check this sentence: “To meet the current market….” because it seems unfinished.

·      Figure 5 should be cited in the text before the figure appears. The same goes Figure 6.

·      In page 12: “When the Miller degree is 20 ~ 70 °CA, both Miller cycles show an upward trend, and the increased range is large.” Unlike the other trends, this trend is not justified.

·      For better view and understanding, would it be possible to add a comparative diagram of miller cycle and conventional diesel engine?

·      It is suggested to add authors’ insight on the effect of duration since miller cycle leads to short intake event that can deteriorate internal aerodynamic loses.

Reviewer 4 Report

In this study, the influence of two Miller cycle technologies on the engine performance and emission characteristics in a diesel engine. This is an interesting topic. I think this article can be accepted after the following minor revision

1.     What are two Miller cycle technologies in this study?

2.     “This paper will analyze B20 concentration biodiesel and focus on the degree of influence on engine power, economy, emission, and combustion characteristics under this concentration ratio [49-50]”. Here, Why should references be added? I think the references here are unnecessary. Besides, what are the combustion characteristics here? General combustion characteristics refer to cylinder pressure, heat release rate and ignition delay.

3.     There are some English grammatical errors in the article.

4.     Some previous studies on reducing diesel exhaust pollution should be added. Effect of injection timing on combustion, emission and particle morphology of an old diesel engine fueled with ternary blends at low idling operations. Energy, 2022, 253: 124150. Comparative study of pilot–main injection timings and diesel/ethanol binary blends on combustion, emission and microstructure of particles emitted from diesel engines. Fuel, 2022, 313: 122658. Optimization of palm oil biodiesel blends and engine operating parameters to improve performance and PM morphology in a common rail direct injection diesel engine. Fuel, 2020, 260: 116326.

Round 2

Reviewer 3 Report

I thank the authors for their response. The manuscript has been revised accordingly, and is recommended for acceptance.

This manuscript is a resubmission of an earlier submission. The following is a list of the peer review reports and author responses from that submission.

Round 1

Reviewer 1 Report

This paper presents performance comparison and optimization of diesel engine fueled with biodiesel based on Miller cycle.

The paper is well-structured. The references are new. 

Figures are visible and understandable. The optimization method is well-described. 

I did not detect any major issues with the paper.

I would suggest only spell/text edit check of the paper and the acceptance of the paper. 

Reviewer 2 Report

The manuscript “Performance Comparison and Optimization of 16V265H Diesel Engine Fueled with Biodiesel Based on Miller Cycle” attempts to report B20 performance in 16V265H diesel engine in two types of Miller cycle, each under different loads.

The findings are useful, but the manuscript is not written well in terms of English, structure, points of discussion, and many more. Based on the overall perspective of how the work is reported, it requires a massive revamp. I can only point out several areas for improvement, but even if they are addressed, an overall re-written manuscript should still be done. In general, the manuscript in incomplete.

1)      It was said that biodiesel can effectively reduce the harmful emissions of diesel engines. This should be supported by data from literature to prove this statement.

2)      Author mentions the necessity to carry out basic research on low fuel ratio biodiesel of locomotive diesel engines. They should define and describe low fuel ratio first.

3)      The statement: “Due to the physical and chemical characteristics of biodiesel itself, its NOx emission will increase after combustion, and the fuel consumption will increase slightly without any change in the structure of fuel injection parameters [48].” is contradicting with an earlier statement: “Based on the unchanged structure of the original whole machine and injection system, biodiesel fuel can effectively reduce the harmful emissions of diesel engines [9].”

4)      The description: “There are two ways to realize Miller’s cycle. One is to change the intake cam profile, keep the intake valve opening time unchanged, and advance the intake valve closing time, that is, change the intake valve lift curve, also known as variable cam profile [15]; The other is to change the installation angle of the intake cam to achieve the purpose of the overlap angle of the variable valve. The opening and closing time of the intake valve will change at the same time, which is called the overlap angle of the variable valve [16-18]. When the diesel engine is burning, the average cylinder temperature of the engine equipped with Miller cycle technology is lower than that of the traditional diesel engine, so the NOx emission is greatly reduced” must be supported by aid of diagram for better visualization.

5)      “Wei et al [22] wanted to study how to reduce the loss of engine fuel economy…”. The word wanted is inappropriate.

6)      Re-write this whole paragraph, as the description is vague and the last part of the last sentence is incomplete: “In this paper, the simulation calculation method is used to study the combustion and emission characteristics, power, and economy of biodiesel [26]. Through the comparison between calculation and test, the performance index of Locomotive Diesel Engine Fueled with biodiesel without structural change is put forward, Optimization Study on the characteristics of Miller cycle of 16V265H diesel engine under three different working conditions when B20 concentration ratio biodiesel is used and the performance optimization of diesel engine by optimizer optimization model [27].” Also, why the need to cite references [26] and [27] in this paragraph? This is not a literature review, but a description of what author is wishing to achieve.

7)      Section 2 needs to be re-written. If it talks about Numerical Approaches, then why the techniques to determine the properties by experimental methods is covered in subsection 2.2?

8)      Subsection 2.1 gives a very lengthy and unnecessary detail description of how general optimization work is done mathematically without clear relation to the results and parameters of what the author is studying. At the moment, this subsection does not carry any values but appears as a literature review of various mathematical methods in optimization process.

9)      Subsection 2.2 should list and cite the standard methods that were used to determine the properties of biodiesel and petrodiesel. The significance or importance of each property, i.e., why they are measured should be discussed in the results and discussion section. Then, author must comment about the values obtained for each property and relate their contribution to the performance of the diesel engine.

10)   Are the values in Table 2 from author’s own work or from the literature?

11)   In this work, the biodiesel was not physically tested on the diesel engine, but rather by qualitative analysis based on the properties of petrodiesel. Therefore, the author should discuss about the properties of biodiesel vs petrodiesel and offer an opinion (with scientific justification supported from literature) of why the performance is better with using biodiesel.

12)   Section 3 is discussing B20 only. Then why did the measurements were also done on other blendings? Also, it appears unclear about the basis of why B20 is the best; was it based on author’s own measurements (If yes, what are they? Need the details), or was it based on deduction made using the data available in the literature (If yes, need the details). This is very confusing.

13)   The two types of Miller cycle need a clear description of working principles. They appear too suddenly in the results and discussion section, without any proper introduction. What are the differences between the two types in terms of principles.

14)   Explain what it means by “load condition”

15)   Fig 5 – 7 can be combined. This applies to Fig 8 – 10, and the rest of the figures. The separation of the figures in its current format is very unnecessary, and it leads to unnecessary lengthy discussions that do not carry new values when moving from one figure to another.

16)   Figure 6 and 7: Trends between the two types of Miller cycle are very similar for loads 75% and 50% respectively. Need to have a clear explanation.

17)   Author wrote: “It can also be seen from the figure that the fuel consumption of the former Miller cycle is slightly greater than that of the latter within this range. When the Miller degree is 20 ~ 70 °CA, both Miller cycles show an upward trend, and the increased range is large. When the Miller degree is gradually greater than 30 ° Ca, the fuel consumption of the former is less than that of the latter.” This is not a standard way of writing. Author must refer specifically to which trend of set of data, instead of representing them with “former/later”.

18)   In all the reasoning/justification of all the trends presented in Section 3, general readers will not be able to understand them without the description of the working principle. It is also wise to use diagram/sketch of the working principle to explain the trend for clear visualization. There’s also a need to cite relevant references when attempting to reason/justify the trends.

Reviewer 3 Report

The paper represents the performance analysis of diesel engine fueled by biodiesel. The submitted paper should be rejected since it has not any scientific value. Moreover, this study has a lot of results and diagrams that are already published in your previous paper in this journal (https://doi.org/10.3390/pr10020372).